# The Impact of Dietary Factors during Pregnancy on the Development of Islet Autoimmunity and Type 1 Diabetes: A Systematic Literature Review

**DOI:** 10.3390/nu15204333

**Published:** 2023-10-11

**Authors:** Valdemar Brimnes Ingemann Johansen, Knud Josefsen, Julie Christine Antvorskov

**Affiliations:** 1Faculty of Health and Medical Sciences, University of Copenhagen, Blegdamsvej 3B, 2200 Copenhagen, Denmark; 2Department of Biology, Faculty of Science, University of Copenhagen, Ole Maaløes Vej 5, 2200 Copenhagen, Denmark; 3Department of Pathology, The Bartholin Institute, Rigshospitalet, Copenhagen Biocenter, Ole Maaløes Vej 5, 2200 Copenhagen, Denmark; knud@eln.dk (K.J.); julie.antvorskov@gmail.com (J.C.A.); 4Steno Diabetes Center, Borgmester Ib Juuls Vej 83, 2730 Herlev, Denmark

**Keywords:** autoantibodies, autoimmunity, childhood diabetes, insulitis, islet autoimmunity, diet, gluten, pregnancy, type 1 diabetes

## Abstract

Aims and hypothesis: The incidence of type 1 diabetes mellitus in children is considerably increasing in western countries. Thus, identification of the environmental determinants involved could ultimately lead to disease prevention. Here, we aimed to systematically review (PROSPERO ID: CRD42022362522) the current evidence of the association between maternal dietary factors during gestation and the risk of developing type 1 diabetes and/or islet autoimmunity (IA) in murine and human offspring. Methods: In accordance with PRISMA guidelines, the present systematic review searched PubMed and Scopus (*n* = 343) for different combinations of MeSH terms, such as type 1 diabetes, diet, islet autoimmunity, prenatal, nutrient, gluten, gliadin, vitamin, milk, and fibers. Results: We found that the most investigated dietary factors in the present literature were gluten, dietary advanced glycosylated end products (dAGEs), vitamin D, fatty acids, and iron. The results concerning prenatal exposure to a gluten-free environment showed a consistently protective effect on the development of IA. Prenatal exposures to vitamin D and certain fatty acids appeared to protect against the development of IA, whereas in utero iron and fat exposures correlated with increased risks of IA. Conclusion: We conclude that a definite association is not established for most factors investigated as the literature represents a heterogeneous pool of data, although fetal exposures to some maternal dietary components, such as gluten, show consistent associations with increased risks of IA. We suggest that human prospective dietary intervention studies in both cohort and clinical settings are crucial to better evaluate critical and protective prenatal exposures from the maternal diet during pregnancy.

## 1. Introduction

Autoimmune diseases result from attacks on the body’s tissues by the immune system and comprise at least 80 diverse diseases where the etiology and pathogenesis are not fully elucidated. The incidence of autoimmunity is approximately 3–5% worldwide, and many of the associated diseases, including type 1 diabetes, are increasing in western societies, especially in children below the age of 5 [1]. Type 1 diabetes is among the most common childhood autoimmune diseases [2]. Considering the geography-dependent rise in incidence as well as the wide variations in disease incidence between neighboring areas, the increase in type 1 diabetes incidence cannot exclusively be explained by genetic changes [3]. Further, from a genetic viewpoint, the high-risk HLA haplotypes and genotypes [4] seem to be less prevalent factors for type 1 diabetes development than previously thought [5,6], underscoring a potentially bigger contribution of environmental determinants. Supporting this environmental or lifestyle-dependent impact of type 1 diabetes etiology, migrants tend to obtain a similar type 1 diabetes risk as the population in their region of relocation [7,8,9]. Additionally, Europeans who are closely related genetically, but separated socioeconomically, tend to acquire substantially different type 1 diabetes risks [10]. Many different environmental factors could contribute to the development of type 1 diabetes such as infections, toxins, and dietary factors [9,11]. The impact of pre- and postnatal environmental factors on disease seems to be crucial because the process leading to IA starts very early in life; seroconversion has a peak incidence or median at around 9–12 months of age in children later diagnosed with type 1 diabetes, illustrating the importance of early environmental determinants [12,13,14].

This systematic review aims to evaluate the existing literature concerning the impact of prenatal, dietary factors as part of the environmental determinants of type 1 diabetes and islet autoimmunity, in murine models and in human case-control and birth cohort studies. We will answer the following questions: (1) Which prenatal dietary factors have already been investigated? (2) Can available data suggest protective advantages or hazards of certain prenatal dietary factors for the development of type 1 diabetes and IA in the offspring? (3) Does the pool of accessible data represent homogenous and conclusive information?

## 2. Research Design and Methods

The systematic review is registered in PROSPERO (CRD42022362522), reported in accordance with Preferred Reporting Items for Systematic review and Meta-Analysis Protocols (PRISMA-P), and follows the PRISMA guidelines, as described extensively in Shamseer et al. and Moher et al. [15,16] to systematically review existing data on the research question (Figure 1A and Appendix A). Our search was performed in October 2022 and included all English-published studies issued until September 2022. The systematic strategy searched for the following medical subject headings terms: type I diabetes, type 1 diabetes, prenatal, diet, islet autoimmunity, nutrient, gluten, gliadin, vitamins, milk, fibers, and animal (complete search strings are available in Appendix A). Combinations were narrowed via the boolean operator “AND”, broadened via “OR” and the use of asterisks allowed our searching strategy to include multiple unknown and variable characters in the search query. The advanced [TITLE-ABS-KEY] and the [ALL FIELDS] functions were used in the cross-disciplinary databases Scopus and PubMed, respectively. The “NOT” boolean was omitted to avoid exclusion of relevant articles. Following filtering of duplicates, relevant articles eligible for full-text review were selected based on the title- and abstract screenings. Potentially relevant articles underwent a full-text evaluation to determine eligibility, and references of both relevant reviews and included articles were reviewed, to possibly identify additional publications that were not disclosed from the database searches (Figure 1A).

The present systematic review assesses the following population, intervention, comparison, and outcome (PICO) elements:Poulation: pregnant murine or human females where data about nutrient factor exposures during prenatal life and the risk of developing type 1 diabetes and/or islet autoimmunity in the offspring are available.Intervention: nutrient interventions and factors that may affect the incidences or occurrences of type 1 diabetes and islet autoimmunity.Comparison: only studies comparing a proper, nonexposed control group with an exposed group were considered eligible.Outcome: changes in type 1 diabetes or IA incidence in offspring of exposed mothers.

In addition to the PICO elements, we anticipate that the identification of prenatal dietary factors that influence the pathogenesis of type 1 diabetes could be used in the prevention of the disease.

### 2.1. Study Design

Reviews and meta-analyses from the search were screened for references, but not included in the final report. Regarding murine studies, only dietary intervention studies were included, whereas both prospective and retrospective population-based human birth cohort and case-control studies were included.

### 2.2. Eligibility Criteria

For murine studies, inclusion criteria were intervention studies, where a defined maternal dietary factor was the exposure, and the readout was either IA or diabetes incidence or time-to-onset assessments in the offspring. Studies where the dietary intervention lasted for only the in utero period and during gestation until weaning were included. As with murine studies, human studies eligible for inclusion had one or more defined maternal dietary factors as prenatal exposure and examined either IA, type 1 diabetes, or both as endpoints. For animal studies examining islet autoimmunity, considered parameters were degree/score/index of insulitis (infiltration of immune cells in the pancreatic islets). Where information of the amount of apoptotic cells, inflammatory cell signaling, inflammatory marker RNA levels, and cytokine protein levels were included in the reviewed articles, this information was also extracted and included in the extensive data tables. Both studies prospectively and retrospectively assessing exposure and outcome were eligible for inclusion. Studies that did not satisfy the stated inclusion criteria were excluded and exclusion reasons were reported (Figure 1A and Appendix A).

### 2.3. Study Selection

An independent reviewer (V. B. I. J.) selected the studies using a two-step screening approach to assess eligibility as described above and in accordance with PRISMA guidelines. The reference lists of selected articles were further searched for relevant articles that were subsequently subjected to the two-step screening method described (Figure 1A). Reviews relevant to the research question allowed relevant publications to be subjected to the two-step screening method following the defined eligibility criteria. J. C. A. and K. J. checked the extracted data.

### 2.4. Data Collection Process

Briefly, data collection concerned study design and outcome (Table 1, Table 2 and Table 3).

## 3. Results

The performed search resulted in 343 articles whereof 99 were duplicates from the two different databases. Following the two-step screening approach, 50 articles were eligible for full-text screening, of which 15 articles were excluded (Figure 1A and Appendix A). More than 25% of the included articles investigated prenatal vitamin D as the exposure, whereas gluten, lipid-related, and iron in utero exposures were examined in 17%, 20%, and 9% of the included articles, respectively (Figure 1B). Other exposures included meat, fish, vegetables, coffee, and tea amongst others (Figure 1C). Importantly, and although their outcomes seemed relevant to the research question, 5 articles [17,18,19,20,21] had not recorded the mothers’ dietary intake and were therefore excluded. These studies were based on serum recordings which were considered as too indirect measures of prenatal exposure, when taking the aim of the present systematic review into consideration. Finally, 35 eligible and qualified articles were selected for final evaluation [22,23,24,25,26,27,28,29,30,31,32,33,34,35,36,37,38,39,40,41,42,43,44,45,46,47,48,49,50,51,52,53,54,55,56], of which 12 studies [22,23,24,25,26,27,28,29,30,31,32,33] were murine and 23 were human [34,35,36,37,38,39,40,41,42,43,44,45,46,47,48,49,50,51,52,53,54,55,56]. All 12 murine studies qualified for the review were dietary intervention studies carried out in either non-obese diabetic (NOD) mice or albino rats (Table 1). Of the 23 included human studies, 8 [34,35,36,37,38,39,40,41] were population-based case-control studies (Table 2), whereas 15 studies [42,43,44,45,46,47,48,49,50,51,52,53,54,55,56] examined the impact of in utero dietary factor exposures on type 1 diabetes and IA risks in population-based birth cohort study designs (Table 3).

### 3.1. Murine Dietary Intervention Studies

All murine studies examined both diabetes and IA as endpoints, except Essien et al. [25] reporting only the impact of maternal nitrosamine-rich charred meat consumption during gestation and lactation on autoimmune diabetes risks in the litters, and McCall et al. [33] reporting on in utero exposure to cadmium and the risk of autoimmune diabetes in the litters (Table 1). The most studied nutrient in the murine dietary intervention studies was gluten [27,29,31], and dietary advanced glycosylated end products (dAGEs) was the second most studied nutrient factor [22,30].

### 3.2. Human Population-Based Studies

Of the 23 eligible human studies, 18 evaluated the association between prenatal, dietary factors and the incidence of type 1 diabetes in the offspring [34,35,36,37,38,39,40,41,45,46,48,49,50,51,52,54,55,56], of which 6 studies additionally reported IA-related readouts [45,46,48,49,55,56]. The first study to improve our understanding of a possible link between prenatal dietary factors and the occurrence of IA was Fronczak et al., [42] and since then, 4 other human population-based birth cohort studies have been focusing entirely on this link [43,44,47,53] (Table 2 and Table 3).

**Table 1 nutrients-15-04333-t001:** The impact of in utero dietary factor exposures to offspring in murine studies.

		Study Design				Outcome		
Reference	Year	Format	Sample Size	Exposure	Readout in Offspring	Findings Described Relative to Control Group	Conclusion	Caution for Interpretation
Peppa et al. [22]	2003	In utero and during lactation dietary intervention study in NOD mice	n_F0_ ^a^ = 8 mating pairs on L-AGE diet, 4 mating pairs on H-AGE diet. 21 of F1 ^b^ and 30 of F2 ^c^ females on L-AGE diet.18 of F1 and 21 of F2 females on H-AGE diet.	Dietary AGEs ^d^ (dAGEs)	IA and autoimmune diabetes	Under LAGE feeding conditions (initiated at 3 or 6 weeks of age), IAA and autoimmune diabetes suppressive effects were significant in F0 mice, and the effects extended throughout F1 and F2 offspring if kept on L-AGE maternal diet (diabetes free rate ~86%). Majority of mice exposed to L-AGE environment maternally or at weaning showed modest insulitis and no autoimmune diabetes >1 year	Perinatally, low dAGE maternal intake lowers the risk of IAA and autoimmune diabetes in the offspring, but no prenatal effect reported	Maternal dietary intervention in perinatal period and not exclusively prenatally Same paternal diet
Hawa et al. [23]	2004	In utero and during lactation dietary intervention study in NOD mice	n_treatment_ = 12 n_control_ = 15	Vitamin D (16 IU in 100 µL olive oil solution)	IA and autoimmune diabetes	No significant differences in autoimmune diabetes incidences, onset of autoimmune diabetes, degree of mononuclear cell infiltration in islets and pancreatic insulin content between the two groups in both diabetic and non-diabetic offspring	The mode and type of vitamin D administration examined here did not prevent autoimmune diabetes in offspring	Dosage of Vitamin D required to achieve effective 1,25-(OH)_2_D_3_ plasma levels could be too low
Arany et al. [24]	2004	In utero and during lactation dietary intervention study in NOD mice	15–20 animals per group (depending on experiment)	Taurine (2.5% *w*/*v*)	IA and autoimmune diabetes	Offspring exposed to taurine had delayed autoimmune diabetes onset, increased survival, decreased autoimmune diabetes conversion rates. Islets from taurine-exposed offspring had almost double the number of PCNA^+^ and IGF-II^+^ cells (*p* < 0.05 and *p* < 0.005), decreased percentages of apoptotic cells (*p* < 0.01), decreased insulitis (*p* < 0.05)	Taurine supplementation administrated in utero and until weaning reduces IAA and delay autoimmune diabetes onset	
Essien et al. [25]	2006	In utero and during lactation dietary intervention study in albino rats	20 female albino rats	Nitrosamine-rich charred meat	Autoimmune diabetes	Dose-related increases and decreases in blood glucose and serum insulin levels, respectively, in litters from pregnant rats that received 40 (*p* < 0.05), 60 and 80% (*p* < 0.01) dosage regimen	Maternal consumption of charred meat leads to hyperglycemia and hyperinsulinemia in the litters	
Kagohashi et al. [26]	2010	In utero and during lactation dietary intervention study in NOD mice	5–14 animals per group (depending on experiment)	*n*-6 and *n*-3 fatty acids (H-chow vs. L-chow) ^e^	IA and autoimmune diabetes	Litters of mothers fed L-chow during gestation and/or lactation had significantly less diabetes (*p* < 0.001). In offspring of mothers fed H-chow during gestation and/or lactation, the severity of insulitis was significantly higher at 6 and 12 weeks of age and there was a trend towards a higher level of IAAs ^f^ at 2 and 4 weeks	*n*-6/*n*-3 ratios of maternal diet in utero and during lactation affects autoimmune diabetes development in NOD mice offspring	When evaluating percentage of diabetic litters, LLH ^g^ should be compared with HHH ^h^ to exclude possible dietary effects from the period after weaning
Hansen et al. [27]	2014	In utero and during lactation dietary intervention study in NOD mice	8–37 per group (depending on experiment)	Gluten	IA and autoimmune diabetes	Feeding a GF ^i^ diet to pregnant mice reduced the cumulative diabetes incidence (*p* < 0.01), increased the onset time (*p* < 0.01) and lowered the insulitis score (*p* < 0.05) in their offspring	A GF diet during gestation and lactation was protective for offspring autoimmune diabetes and IA	
Wang et al. [28]	2014	In utero and during lactation dietary intervention study in NOD mice	10 or 20 per group (depending on experiment)	High fat diet (HFD)	IA and autoimmune diabetes	Female offspring exposed to maternal HFD ^j^ showed more severe lymphocyte infiltration (*p* < 0.05), elevated inflammatory NF-KB signaling (*p* < 0.01), TNFa protein level (*p* < 0.05) and a trend towards decreased p-Akt (*p* < 0.1) in pancreatic tissue, impaired glucose tolerance (*p* < 0.05) and lower serum insulin levels (*p* < 0.05)	Maternal HFD accelerates the autoimmune diabetes development in offspring NOD mice.	
Antvorskov et al. [29]	2016	In utero and during lactation dietary intervention study in NOD mice	15–24 per group (depending on experiment)	Gluten	IA and autoimmune diabetes	NOD mice GF in utero showed the lowest diabetes incidence (*p* < 0.0001), less lymphocytic infiltration (*p* < 0.001), reduced Th17 cell characteristic RORyt ^k^ expression level (*p* < 0.0001)	GF diet exclusively during pregnancy almost completely prevented autoimmune diabetes in offspring	
Borg et el. [30]	2018	Perinatal dietary intervention study in NOD mice	5–10 per group (depending on experiment)	dAGEs (Low (LAGE) vs. high (HAGE))	IA and autoimmune diabetes	NOD mice exposed to LAGE from conception to early postnatal life had reduced insulitis (*p* < 0.001), similar overall insulitis index and no difference in islet number or islet area. The islets from those mice had 4-fold higher basal (*p* < 0.002) and glucose-stimulated (*p* < 0.02) insulin secretion, similar stimulation index and total insulin content and 4-fold higher basal and glucose stimulated proinsulin secretion (*p* < 0.002).	Reducing exposure to dietary AGEs during gestation, lactation, and early postnatal benefits insulin secretion and insulitis.	The diet intervention lasted from conception until post-natal day 28 making it difficult to know how the gestational feeding period per se contributes phenotypically
Haupt-Jørgensen et al. [31]	2018	In utero dietary intervention study in NOD mice	2–15 (depending on experiment)	Gluten	IA and autoimmune diabetes	NOD mice GF in utero showed decreased insulitis (*p* < 0.05) at 13-weeks but not at 4 weeks of age, no significant difference in TG ^l^ activity in islets, lower expression of IFNγ ^m^ in total splenic cells (*p* < 0.05) and in splenic CD4^+^CD3^+^ T cells (*p* < 0.0001), lower expression of IL22 in ydTCR^+^CD3^+^ T cells (*p* < 0.05), no differences in the overall proportions of analyzed T cells from the lymphoid organs, increased total islet number (*p* < 0.01), but showed no differences in total beta-cell volume, total islet volume or total pancreas volumes.	A GF in utero milieu reduced insulitis at 13 weeks of age	The TG activity assay only reported about activity in islets from 13-week-old mice, which does not rule out a significant difference at an earlier pre-diabetic state before severe IA.
Huang et al. [32]	2018	Perinatal dietary intervention study in NOD mice	4–36 (depending on experiment)	Dietary GEN ^n^	IA and autoimmune diabetes	Perinatally GEN-dosed female offspring showed earlier autoimmune diabetes onset and higher incidence rate (*p* < 0.05 at PND ^o^ 113, 120, 137, 162) and significant changes in the inflammatory environment as well as shifting towards a pro-inflammatory response, which cooccurred with the accelerated incidence rate of autoimmune diabetes	Female litters being perinatally expose to GEN showed accelerated autoimmune diabetes phenotypes, but no causal effect of GEN-induced changes regarding IA is reported	The diet intervention lasted from embryonic day 7 to postnatal day 21
McCall et al. [33]	2021	In utero dietary intervention study in NOD mice	4–34 (depending on experiment)	Cadmium	Autoimmune diabetes	Prenatally cadmium-exposed NOD mice had similar time-to-onset of autoimmune diabetes, similar autoimmune diabetes incidence, no changed nTreg percentages, total nTreg cell numbers or any other major splenocytic phenotypes. No biologically significant differences in the splenic cytokine production were observed	Prenatal cadmium exposure did not alter the development of autoimmune diabetes	The study is not assessing IFNG or anti-inflammatory cytokines

^a^ Founder generation. ^b^ Generation one. ^c^ Generation two. ^d^ Advanced Glycosylated End-products. ^e^ Chow diets containing *n*-6/*n*-3 fatty acid ratios of 14.5 and 3.0, respectively. ^f^ Insulin autoantibody. ^g^ L-chow fed during gestation and lactation, but H-chow fed after weaning. ^h^ H-chow fed during gestation, lactation, and after weaning. ^i^ Gluten free diet. ^j^ 26.2% protein, 26.3% carbohydrates, 34.9% fat with 60% energy from fat. ^k^ Retinoic acid receptor-related orphan receptor *γ* t (a Th17 characteristic nuclear transcription factor). ^l^ Transglutaminase. ^m^ Interferon *γ*. ^n^ Isoflavone genistein. ^o^ Post natal day.

**Table 2 nutrients-15-04333-t002:** The impact of in utero dietary factor exposures to offspring in human population-based case-control studies.

		Study Design							
		Sample		Exposure		Readout		Outcome	
Reference	Year	Size (*n*)	Subject Type	Nutrient Factor	Assessment	Type of Endpoint	Assessment	Findings Described Relative to Control Group	Conclusion
Virtanen et al. [34]	1994	600 cases 536 randomly selected controls	Diabetic Finnish children younger than 15 years old diagnosed from September 1986 to April 1989	Coffee and tea	Retrospectively via FFQ ^a^	Type 1 diabetes	Retrospectively	Maternal coffee or tea consumption during pregnancy did not affect the risk for diabetes in the children	No observed increased risk for type 1 diabetes in children whose mothers drank coffee or tea during pregnancy
Stene et al. [35]	2000	85 cases 1071 random controls	Both groups from Vest-Agder, Norway	Cod liver oil and vitamin D supplements	Retrospectively via FFQ	Type 1 diabetes	Retrospectively	Offspring exposed in utero to cod liver oil had a lower risk of developing type 1 diabetes (OR = 0.30, 95% CI: 0.12–0.75, *p* = 0.01) whereas exposure to vitamin D supplements did not (OR = 1.11, 95% CI: 0.69–1.77)	Consumption of cod liver oil during gestation was associated with reduced risk of type I diabetes in the offspring
Stene et al. [36]	2003	545 cases 1668 random controls	Norwegian children diagnosed between 1997 and 2000 and born between 1985 and 1999	Cod liver oil and vitamin D supplements	Retrospectively via FFQ	Type 1 diabetes	Retrospectively	No clear association between maternal use of cod liver oil or other vitamin D-containing supplements during pregnancy and type 1 diabetes among children	In utero exposure to cod liver oil is not associated with the risk of type 1 diabetes
Sipetic et al. [37]	2005	105 cases 210 outpatient controls	Belgradian children born 1994–1997 and ≤16 years old	Nitrosamines-rich food, coffee, tea, coca cola, alcohol	Retrospectively via FFQ 6–12 weeks after diagnoses with type 1 diabetes	Type 1 diabetes	Retrospectively	Mothers to type 1 diabetes offspring consumed more coffee (*p* < 0.001), coca cola (*p* = 0.001), alcohol (*p* < 0.001) and nitrosamines-rich foods (*p* < 0.0001) according to conditional univariate logistic regression. Maternal consumption of nitrosamines-rich foods during pregnancy was associated with increased type 1 diabetes risk in offspring univariately (OR = 5.96, 95% CI: 2.76–12.84) and multivariately (OR = 4.33, 95% CI: 1.95–9.61, *p* < 0.001)	Maternal consumption of nitrosamines-rich foods during pregnancy was independently related to type 1 diabetes in the offspring
Muntoni et al. [38]	2013	123 cases 127 controls	Sardinian children of 0–15 years old	Vegetables, meat, lipids, fruits, fish, dairy products, cereals, carbohydrates, beverages, and alcohol	Retrospectively via FFQ	Type 1 diabetes	Retrospectively	None of the examined food variables showed a significant association to type 1 diabetes, when evaluating the maternal diet under pregnancy. A trend toward significance (*p* = 0.059) was observed for the category of meat consumption during pregnancy and lactation	No associations between type 1 diabetes in the offspring and the investigated dietary factors in the maternal diet under pregnancy
Jacobsen et al. [39]	2015	127,207 exposed children 69,667 medium-exposed children 134,749 controls	All individuals born in Denmark from 1983–1988	Margarine fortified with vitamin D	Retrospectively (defined by subject group)	Type 1 diabetes	Retrospectively through the Danish Childhood Diabetes Registry	The beta coefficients (calculated as slopes) for linear increase in the risk of type 1 diabetes until age 15 years after adjustments were 0.007 (*p* > 0.001) and did not differ between the 3 gestational exposure groups, suggesting that the type 1 diabetes risk rose steadily without margarine fortification changing the pattern	No evidence that exposure to low vitamin D doses from the margarine fortification in utero is changing the risk of developing type 1 diabetes
Thorsen et al. [40]	2019	257 cases ~260 controls/case	Children from pregnant Danish women from 1996–2002	Iron supplementation	Retrospectively via FFQ data	Type 1 diabetes	Retrospectively through registry (prospectively in the register)	Maternal pure iron supplementation during pregnancy was not associated with later risk of offspring type 1 diabetes (HR = 1.05, 95% CI: 0.76–1.45). This held true for when comparing offspring risks prior to or after gestational week 20 (HR = 0.82 and 1.13, 95% CI: 0.57–1.17 and 0.83–1.53, respectively)	Prenatal iron exposure through supplementation did not lead to a higher risk of childhood type 1 diabetes.
Pazzagli et al. [41]	2021	1654 cases 779,913 controls	Singleton pregnancies between 2005 and 2018	Folic Acid	Prospectively	Type 1 diabetes	Prospectively	Children with prenatal folic acid exposure had similar odds for developing neonatal diabetes or hyperglycemia (OR = 0.95, 95% CI: 0.72–1.25) and for developing type 1 diabetes (HR = 1.05, 95% CI: 0.93–1.18).	Risks of developing neonatal diabetes, hyperglycemia or type 1 diabetes were not associated with prenatal folic acid exposure

^a^ Food frequency questionaries.

**Table 3 nutrients-15-04333-t003:** The impact of in utero dietary factor exposures to offspring in human population-based birth cohort studies.

		Study Design							
		Sample		Exposure		Readout		Outcome	
Reference	Year	Size (*n*)	Subject Type	Nutrient Factor	Assessment	Type of Endpoint	Assessment	Findings Described Relative to Control Group	Conclusion
Fronczak et al. [42]	2003	16 cases 206 controls	DAISY ^a^ children	Vitamin D via food and via supplements, *n*-3 and *n*-6 fatty acids	Retrospectively via Willet FFQ	IA	Prospectively (range 0.8–7.3 years)	Maternal intake of vitamin D through food was associated with a decreased risk of IA in offspring univariately (adjusted HR = 0.37, 95% CI: 0.17–0.78)	In utero exposure to vitamin D through food may have a protective effect on the appearance of GAD_65_, IA-2 and insulin autoantibodies
Brekke et al. [43]	2007	11,081 at 1 year 8805 at 2.5 year	ABIS ^b^ children	Vitamin D supplementation	Retrospectively via FFQ after birth	IA	Prospectively	Maternal consumption of vitamin-D-containing supplements during pregnancy was negatively associated with the appearances of diabetes-related autoantibodies ^e^ in 1 year offspring (OR = 0.708, 95% CI: 0.520–0.964, *p* = 0.028), but this association was not seen in 2.5-year offspring	Use of vitamin-D-containing supplements during pregnancy was associated with reduced IA in offspring at 1 year but not at 2.5 year of age
Lamb et al. [44]	2008	642 in analysis cohort	DAISY children	Potatoes, other root vegetables, gluten-containing foods, non-gluten cereal grains, cow’s milk products, fruits, vegetables, meat, fish	Retrospectively via Willet FFQ	IA	Prospectively (0.8–15 years)	Maternal increased frequency of potato consumption during last trimester of pregnancy was associated with a delayed IA onset (HR = 0.58, 95% CI: 0.34–1.01). Earlier onset of IA in the offspring was further associated with lower potato consumption in the fully adjusted model (HR: 0.49, 95% CI: 0.28–0.87)	No association between maternal frequency of consumption of other root vegetables, gluten-containing foods, non-gluten cereal grains, cow’s milk products, fruits, vegetables, meat or fish during pregnancy and the IA onset time, however potato consumption was
Uusitalo et al. [45]	2008	3727 children in analysis cohort	DIPP ^c^ children	Retinol, β-Carotene, Vitamin C, Vitamin E, Selenium, Zinc, Manganese	Retrospectively via FFQ (1–3 months after birth)	IA and type 1 diabetes	Prospectively (observed every 3–12 months)	Maternal intake of the antioxidant nutrients analyzed showed no significant associations with the risk of IA in the offspring during the median follow-up time (4.4 years) (HR close to one, Cis tended to be wider than in main analysis due to lower number of cases)	None of the maternal dietary antioxidant intakes were associated with the risk of advanced β-cell autoimmunity
Marjamäki et al. [46]	2010	3727 children in analysis cohort	DIPP children	Vitamin D from foods and supplements	Retrospectively via FFQ (1–3 months after birth)	IA and type 1 diabetes	Prospectively (observed every 3–12 months)	Neither of the In utero exposures to vitamin D from food, supplements or combined sources were associated with advanced β-cell autoimmunity/clinical type 1 diabetes in the offspring, when adjusting for genetic risk and familial type 1 diabetes	HLA-conferred susceptible DIPP children don’t have a higher risk of developing IA when exposed to vitamin D from food or supplements in utero
Brekke et al. [47]	2010	5724 in analysis cohort	ABIS children	Potatoes/root vegetables, fried potatoes/french fries, chips, vegetables, cream/crème fraiche, meat (cow, calf, ox), meat/sausage (pig), meat (wild animals), fish (open sea), fish (lake), fish (Baltic sea), eggs, pastries, candy (non-chocolate), chocolate, mushroom (field), milk/sour milk/yoghurt, slices of bread, coffee	Retrospectively via FFQ after birth	IA	Prospectively	Less than maternal daily consumption of vegetables was associated univariately with increased risk of IA in the child (OR = 1.71, 95% CI: 1.24–2.35, *p* = 0.001) and the association persisted when combining the three lower frequency categories and compared to daily vegetable consumption. The association was strengthened when adjusting for known IA-risk factors (*p* for trend <0.001). No other food tended to associate with the risk of IA except less than daily consumption of coffee which tended to associate with decreased risk of IA (*p* for trend = 0.014)	Daily vegetable during pregnancy might protect against IA risk in the offspring
Virtanen et al. [48]	2011	3727 children in analysis cohort	DIPP children	Milk and milk products, cereal products, meat and meat products, fish/fish products/shellfish, eggs, dietary fats, vegetables, roots and potatoes, fruits, berries, fruit and berry juices, chocolate and sweets, use of alcoholic drinks, tea, coffee	Retrospectively via FFQ (1–3 months after birth)	IA and type 1 diabetes	Prospectively (observed every 3–12 months)	In utero exposure to butter. low-fat margarines, berries (HR = 0.83, 0.58, 0.92, 95% CI: 0.70–0.98, 0.38–0.89, 0.85–1.00 respectively) and coffee (overall significance for pooled coffee variable was *p* = 0.127) were inversely associated univariately with IA when adjusting for genetic risk and familial diabetes. All associations except that for butter remained significant when all those foods were included in the same model. The rest of the tested food categories showed no associations	Maternal intake of butter, low-fat margarine, berries and coffee during pregnancy showed weak associations with advanced IA in offspring determined by ICA, IAA, GADA and IA-2A levels
Niinisto et al. [49]	2014	4887 children in analysis cohort	DIPP children	Fatty acids (SFA, MUFA, PUFA, conjugated linoleic acid) and cow’s milk products, fresh milk, cheese, sour milk, butter and butter-oil spreads, low-fat margarines, high-fat margarines, oil, red meat and meat products, poultry, fatty fish, lean fish	Retrospectively via FFQ (1–3 months after birth)	IA and type 1 diabetes	Prospectively (observed every 3–12 months until 0.5–11.5 years)	Maternal intake of saturated palmitic acid was weakly associated with a decreased risk of type 1 diabetes (HR = 0.82, 95% CI: 0.67–0.99, *p* = 0.039). Considering clinical type 1 diabetes as an endpoint, onset time and consumption of the following fatty acids were significantly associated: palmitoleic acid isomers 16:1n-7 (*p* = 0.019) and 16:1n-9 (*p* = 0.014), EPA (*p* = 0.037), dihomo-γ-linolenic acid (*p* = 0.013). High cheese, low-fat margarines consumption associated with a decreased risk of type 1 diabetes (HR = 0.52 and 0.67, 95% CI: 0.31–0.87 and 0.49–0.92 respectively) and high consumption of sour milk, fat from fresh milk and protein from sour milk with increased IA or type 1 diabetes (HR = 1.14, 1.43 and 1.15, 95% CI: 1.02–1.28, 1.04–1.96 and 1.02–1.29 respectively)	Some of the maternal dietary components investigated during gestation are weakly associated with IA and type 1 diabetes development in the offspring
Granfors et al. [50]	2016	16,339 children in analysis cohort	ABIS babies	Vitamin D-containing mineral and multivitamin supplements	Retrospectively via FFQ after birth	Type 1 diabetes	Prospectively	Vitamin D supplement use during gestation was reported by 9.3% of mothers whose children later developed type 1 diabetes and among 11.3% of those mothers who had nondiabetic children at 14–16 years of age (*p* = 0.532)	No significant association was found between reported maternal intake of vitamin D-containing supplements and risk of type 1 diabetes development in the child
Størdal et al. [51]	2018	94,209 children	MoBa ^d^ babies	Iron supplementation	Retrospectively	Type 1 diabetes	Prospectively	The incidence rate of type 1 diabetes among children exposed to iron supplementation in utero was higher after adjustment (HR = 1.33, 95% CI: 1.06–1.67) and this was still the case when mothers only supplementing with either iron-only or iron-other supplements were analyzed alone	Prenatal iron exposure led to a higher risk of type 1 diabetes
Antvorskov et al. [52]	2018	67,565 children	Children from Danish women pregnant in the period between 1996 and 2002	Gluten	Retrospectively via FFQ	Type 1 diabetes	Retrospectively through registry linkage but followed prospectively	Comparing offspring of mothers with highest gluten intake with those of mothers with lowest gluten intake, offspring had double the risk of type 1 diabetes development later in life (HR = 2.00, 95% CI: 1.02–4.00). Offspring type 1 diabetes risk was positively associated with prenatal exposure (*p* trend = 0.016). This association persisted after expressing it as energy adjusted residuals (*p* trend = 0.028) and after describing gluten intake as groups of 20% (*p* trend = 0.035)	In utero gluten exposure was strongly associated with the risk of type 1 diabetes later in life
Silvis et al. [53]	2019	8676 children	TEDDY ^e^ children	Intake of supplemental vitamin D and *n*-3 FAs	Retrospectively via FFQ postpartum	IA	Prospectively (every 3 month between 3 and 48 months of age, subsequently every 6 months)	Maternal intake of vitamin D supplements was not associated with the risk of IA (HR = 1.11, 95% CI: 0.94–1.31), IAA-first IA (HR = 1.24, 95% CI: 0.94–1.62) or GADA-first IA (HR = 1.01, 95% CI: 0.79–1.29) Maternal intake of *n*-3 FA supplements was not associated with the risk of IA (HR = 1.19, 95% CI: 0.98–1.45), IAA-first IA (HR = 1.22, 95% CI: 0.89–1.68) or GADA-first IA (HR = 1.27, 95% CI: 0.95–1.70)	Maternal use of vitamin D and *n*-3 FA supplements during pregnancy and risk for offspring IA was not associated
Lund-Blix et al. [54]	2020	86,306 children	MoBa children	Gluten	Retrospectively via FFQ	Type 1 diabetes	Retrospectively through registry (prospectively in the register)	The global likelihood ratio test showed no linear association between increases in gluten intake and type 1 diabetes childhood risk (*p* = 0.11). For each 10 g/day-increase of gluten intake, adjusted HR was 1.02 (95% CI: 0.73–1.43, *p* = 0.91), and for each standard deviation increase of gluten intake, adjusted HR was 1.01 (95% CI: 0.85–1.20, *p* = 0.91)	Maternal gluten intake in pregnancy was not associated with offspring type 1 diabetes risk
Mattila et al. [55]	2020	4879 children in analysis cohort	DIPP children	Nitrate and nitrite from food groups	Retrospectively via FFQ (1–3 months postpartum)	IA and type 1 diabetes	Prospectively (observed every 3–12 months during 15-y follow-up)	Adjustments through two different models both led to no association between prenatal exposure to nitrate and nitrite from diet (HRs ranging from 0.97–1.03, 95% CI ranging from 0.83–1.12 to 0.92–1.16)	Neither maternal intake of nitrate or nitrite from diet during pregnancy showed an association to development of IA or type 1 diabetes in the offspring
Mattila et al. [56]	2021	4879 children in analysis cohort	DIPP children	Vitamin C and iron	Retrospectively via FFQ (1–3 months postpartum)	IA and type 1 diabetes	Prospectively (observed every 3–12 months during 15-y follow-up)	Maternal use of vitamin C supplements during pregnancy was not associated with offspring IA risk (HR = 1.08, 95% CI: 0.86–1.33, *p* = 0.51) or type 1 diabetes risk (HR = 1.18, 95% CI: 0.84–1.58, *p* = 0.43). Maternal use of iron supplements during pregnancy was not associated with offspring IA risk (HR = 1.18, 95% CI: 0.91–1.51, *p* = 0.21) or type 1 diabetes risk (HR = 1.17, 95% CI: 0.84–1.62, *p* = 0.36). Similar results seen from total intake (supplements + intake from diet)	Neither maternal intake of vitamin C or iron from vitamin C and iron only supplements as well as multivitamin supplements during pregnancy showed an association to development of IA or type 1 diabetes in the offspring

^a^ The Diabetes Autoimmunity Study in the Young. ^b^ All Babies in Southeast Sweden. ^c^ The Type 1 Diabetes Predicition and Prevention Study. ^d^ The Norwegian Mother, Father and Child Cohort Study. ^e^ The Environmental Determinants of Diabetes in the Young.

## 4. Discussion

### 4.1. Vitamin D

Low vitamin D levels have been linked to the development of autoimmune diseases [57], possibly due to its impact on both the innate and adaptive immune systems [58,59]. Data from both animal and human studies showed conflicting observations in terms of prenatal vitamin D exposure and the type 1 diabetes risk during early life. In a study performed by Hawa et al. [23], maternal administration of olive oil containing vitamin D showed no significant effect on type 1 diabetes incidence, onset time, and IA when comparing NOD mice offspring from control and exposure groups. However, early life administration of vitamin D has also shown protective effects in animal studies [60,61].

In 2003, Fronczak and colleagues reported that maternal vitamin D intake during pregnancy had a protective effect on the child’s risk of developing IA by evaluating measures of GAD_65_, IA-2, and insulin autoantibodies in children from the Diabetes Auto Immunity Study in the Young (DAISY) study [42]. Interestingly, a similar effect was reported in children from the All Babies in Southeast Sweden (ABIS) study, a large, prospective, population-based cohort study of children born during 1997–1999. Here, maternal vitamin D supplementation during pregnancy protected against IA, when evaluating offspring at 1 year of age [43]. However, in a Finnish type 1 Diabetes Prediction and Prevention (DIPP) prospective birth cohort, including children genetically at risk of type 1 diabetes born between 1997–2004, neither in utero exposures to vitamin D from food, supplements, or in combination could support the findings of advanced β-cell autoimmunity from the DAISY and ABIS Children [46]. Also, when evaluating the ABIS children cohort 9 years after the first publication, Granfors and colleagues [50] could not support the protective role of vitamin D from 2007 [43], when they analyzed the in utero vitamin D exposure as a continuous and binary variable retrospectively, and when assessing the IA-risk in Finnish children with increased genetic risk for type 1 diabetes. The Environmental Determinants of Diabetes in the Young (TEDDY) study found no association between the intake of Vitamin D during pregnancy and the risk of developing IA in the offspring [53]. Although they show inconsistencies in their findings, the obvious strengths of the prospective DAISY, ABIS, DIPP and TEDDY studies are the exactness of data collection concerning confounders and endpoints. However, the accounts for exposures were obtained retrospectively in those studies.

In the early 2000s, a Norwegian research team studied the effect of administrating cod liver oil and other vitamin D-containing supplements during pregnancy. They found that consumption of cod liver oil during pregnancy had a protective effect on the risk of offspring type 1 diabetes [35] whereas other vitamin D-containing supplements, such as maternal multivitamin supplementation, showed no significant effects. Assuming that vitamin D in cod liver oil was responsible for the observed effect, they argued that the bioavailability of vitamin D from cod liver oil might be better than that of vitamin D from multivitamin supplements as a possible explanation for their observations [35]. It is, however, still a possibility that the negative association between offspring type 1 diabetes incidence and maternal cod liver oil intake came from other cod liver oil constituents. This publication was a retrospective, human, population-based case-control study design with people from a defined region in Norway as the study population [35]. The research group extended their study with cases and controls covering a much broader area of Norway 3 years later [36]. Importantly, they saw no significant, protective effects of administering cod liver oil and other vitamin D-containing supplements in this extended study [36]. A Danish study, including data from 331,623 children, used vitamin D-fortified margarine to assess a potential effect on the development of type 1 diabetes when exposed in utero, but they also did not identify any association between maternal vitamin D supplement and infant type 1 diabetes risks [39].

### 4.2. Gluten

Proline and glutamine-rich gluten proteins are highly hydrophobic and partially resistant to intestinal degradation, which makes them more immunogenic than other nutritional proteins that are efficiently hydrolyzed into single amino acids [52]. Markedly, all animal studies examining a maternal gluten-free (GF) diet as the dietary intervention consistently showed that a prenatal, GF environment protected against autoimmune diabetes and IA. As demonstrated by Antvorskov et al. [29], offspring of NOD mice, fed a GF diet only during pregnancy, had a significantly reduced diabetes incidence (from 8.3% in mice, where the mother was GF specifically during pregnancy, to 62.5% in offspring of mothers, eating a gluten-containing diet during pregnancy). Pups from mothers, fed a GF diet, also showed a significantly lower insulitis score in the same study [29]. In mice, that were not GF in utero, this correlated with increased expression of RORγt, a nuclear transcription factor characteristic for Th17 cells, that was significantly changed by diet. These findings were supported by Hansen et al. who found that the intestinal gene expression profile was skewed towards an anti-inflammatory phenotype in offspring of GF diet-fed mice [27]. Although not detected in the present systematic literature review, a recent study also found that the insulitis score in GF mice was significantly reduced compared with mice fed a standard diet, and that the markers for regulatory T cells and T helper 2 cells were upregulated in the pancreas of GF mice [62].

The human DAISY study revealed that maternal consumption of neither gluten-containing foods nor non-gluten cereal grains during pregnancy was associated with the incidence or onset of IA in the children [44]. In the DIPP cohort with 3727 children in 2011, Virtanen and colleagues found no association with advanced IA in children from mothers consuming cereal products [48]. However, results from a large Danish National Birth Cohort Study [52], comprising 101,042 pregnancies, were published in BMJ in 2018. In this study, maternal gluten intake, based on maternal consumption of gluten-containing foods, was reported from a food frequency questionnaire during pregnancy. The average gluten intake was 13.0 g/day, ranging from less than 7 g/day to more than 20 g/day. The incidence of type 1 diabetes in the participants’ children was 0.37% (*n* = 247) with a mean follow-up period of 15.6 years, and the risk of type 1 diabetes in the children increased proportionally with maternal gluten intake during pregnancy. Women with the highest gluten intake doubled the risk of type 1 diabetes development in their offspring, which illustrates a potential for gluten intake during pregnancy to significantly affect the childrens’ risk for type 1 diabetes development, supporting the animal studies. However, the findings were not confirmed in the Norwegian birth cohort study, the MoBa cohort [54]. The difference between the two studies could be due to the different gluten content in the wheat, rye, and barley used for manufacturing and consumption in the two different countries [52].

### 4.3. Fatty Acids

Evaluation of NOD mice offspring from 0–50 weeks after birth showed that offspring, born from and lactated by mothers fed a chow diet with high *n*-6/*n*-3 fatty acid ratio during the gestation and/or lactation period, developed diabetes faster than offspring from mothers, who were fed a chow diet with low ratio during pregnancy. Interestingly, pups fed a high-fat chow diet after weaning, but born from mothers fed a low-fat chow diet, had a significantly lower diabetes incidence compared to the group of pups fed a low-fat chow diet, but born from mothers fed a high-fat chow diet during gestation and/or lactation [26]. Additionally, a maternal high-fat diet accelerated autoimmune diabetes development in offspring from NOD mice [28]. Similarly, among 4887 children in the DIPP study, maternal intake of certain fatty acids, including EPA, was associated with a decreased risk of type 1 diabetes [49]. However, in the DAISY study, no significant association was found between a combined intake of EPA and DHA and the risk of IA in the children [42], and in the TEDDY cohort, maternal consumption of *n*-3 fatty acid supplementation during pregnancy was not associated with the risk of IA in the offspring [53].

### 4.4. Iron Supplementation

A case-cohort study, using the Danish National Birth Cohort comprising approximately 100,000 Danish pregnant women, and CPR linkage to the Danish Childhood Diabetes Register, found no association between maternal consumption of iron during pregnancy and type 1 diabetes in the offspring [40]. Iron overload can, however, lead to the formation of reactive oxygen species (ROS) in pancreatic islet cells which possess a classical iron metabolism and are sensitive to ROSs [63,64,65]. In contrast, in the Norwegian MoBa study, it was found that prenatal exposure to iron could be a risk factor for type 1 diabetes [51]. Importantly, maternal, but not fetal, human homeostatic iron regulator protein (HFE) genotypes, causing specific iron storage levels, were associated with offspring type 1 diabetes [51], potentially underlining the maternal dependency. Further, a very recent study focusing on the DIPP children found no association between IA or type 1 diabetes risks in children, exposed to iron in utero by maternal supplementation, when compared to nonexposed children [56].

## 5. Conclusions

In utero dietary deficiencies or overloads might alone, or in combination with other environmental factors, play a role in the process leading to the development of type 1 diabetes and IA in children. A potential mechanism of how diet during pregnancy could influence the development of IA and type 1 diabetes in the offspring could be the promotion of either an inflammatory or anti-inflammatory environment [29,31,62,66,67] affecting the child’s developing immune system. Furthermore, maternal nutrition during pregnancy can potentially induce epigenetic changes in the child [68] or influence the composition of the resident microbiota that is transferred to the child during a vaginal birth. 

This systematic review followed the PRISMA guidelines and evaluated the present literature focusing on in utero dietary exposures and the impact on type 1 diabetes and IA. Maternal consumption of nutritional products, such as gluten, iron, and high fat diet during pregnancy was associated with increased risks of type 1 diabetes and/or IA in the offspring, whereas in utero vitamin D and specific fatty acid exposures protected against these endpoints in some studies. However, these results represent a rather heterogeneous pool of data, and inconsistencies are present, illustrating the importance of research in this field. No prenatal, dietary factor has unequivocally been identified that modulate the frequency of T1D in the child. Thus, it is at present difficult to recommend or discourage specific components from the maternal diet to minimize the disease development in a child with high genetic risk for T1D. Our findings should initiate novel human, prospective, dietary intervention studies in both cohort and clinical settings, which are needed to better understand critical and protective prenatal exposures from the maternal diet during pregnancy. Conclusions from such studies could be an important way to prevent the development of type 1 diabetes in children at genetic risk of the disease.

A contributing factor that may be responsible for the lack of consistency in the current literature is that we likely do not know all the risk factors of IA and type 1 diabetes at present. This would explain the heterogenous results and it suggests basic research within this field to further elucidate the environmental and genetic determinants underlying type 1 diabetes. Importantly, committees, guiding and advising pregnant women concerning prenatal dietary exposure, need to carefully consider the findings in the literature. Furthermore, there is a lack of a concluding studies and review/meta-analysis investigating the effect of non-medical interventions (for example, the role of psychical activity, diet, probiotics, sugar-containing beverages, psychosocial support) that could be used in IA positive children to prevent progression to type 1 diabetes.

## Figures and Tables

**Figure 1 nutrients-15-04333-f001:**
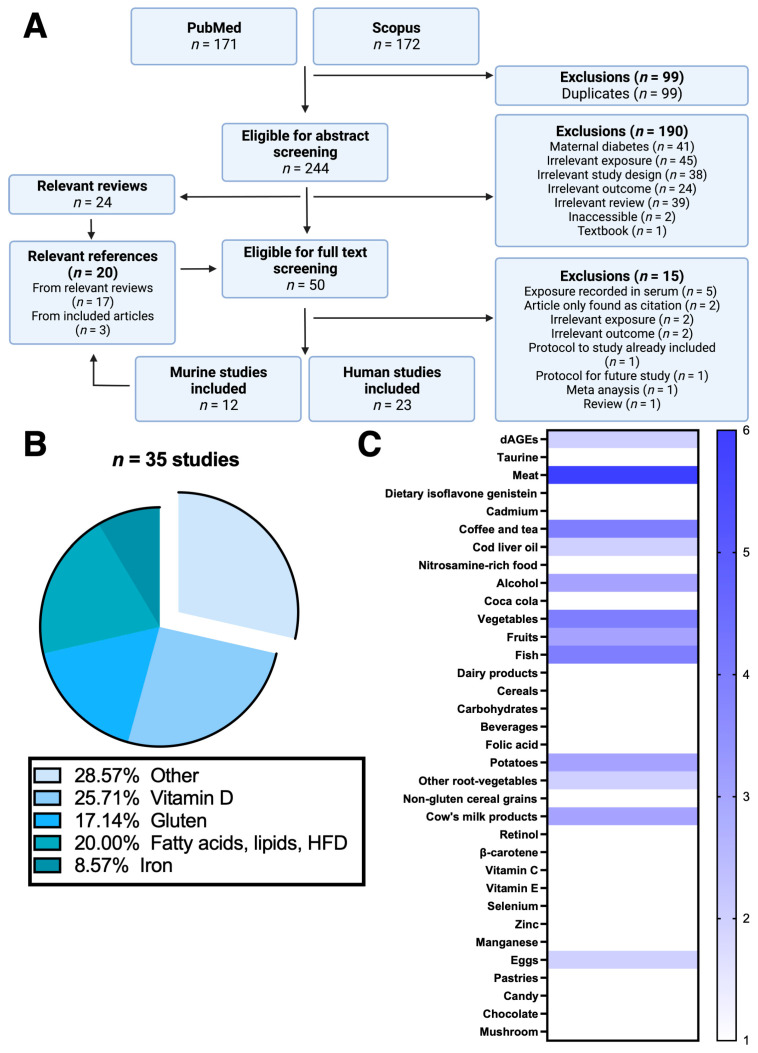
PRISMA flow chart and summary of data extracted. (**A**) Flow chart of the systematic literature review and study selection process by following the PRISMA guidelines. Exclusion reasons summarized to the right. Find official PRISMA flow chart in Appendix A. (**B**) Pie chart of the included studies from (**A**) showing the proportion of the investigated dietary factors during pregnancy. Dietary factors investigated in the exploded category named “other” are shown in (**C**) in a heatmap format, where each dietary factor is scored according to how many times it has been investigated in the included articles from the “other” category. High-fat diet (HFD).

## Data Availability

Data extraction from literature screening is available in the included figures of the present manuscript, in PROSPERO as well as in the Appendix A.

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
