# Peer review of "The Impact of Dietary Factors during Pregnancy on the Development of Islet Autoimmunity and Type 1 Diabetes: A Systematic Literature Review"

_nutrients, 2023, doi:10.3390/nu15204333_

Round 1

Reviewer 1 Report

the purpose is very interesting. methods are correct. but the results have been presented in an unacceptable way: the tables are illegible and the results little understandable. I suggest that 1) data must be synthetized even with help of some signs (arrow up or down?), 2) smaller tables could be made dedicated to just one nutrient (gluten, vit D, cadmio, others), also because the details of each study has been presented in "discussion". I can add that Authors could be clarify for which nutrients it is possible to conclude something and for which no.

Author Response

Reviewer 1

The purpose is very interesting. methods are correct. but the results have been presented in an unacceptable way: the tables are illegible and the results little understandable. I suggest that 1) data must be synthetized even with help of some signs (arrow up or down?), 2) smaller tables could be made dedicated to just one nutrient (gluten, vit D, cadmio, others), also because the details of each study has been presented in "discussion". I can add that Authors could be clarify for which nutrients it is possible to conclude something and for which no.

Response: Thank you very much for the comments. We agree that our results synthesized in tables are comprehensive and contain a lot of information. The tables are made in accordance with the guidelines of the Preferred Reporting Items for Systematic review and Meta-Analysis Protocols (PRISMA-P) and PROSPERO (please refer to our supplementary materials) that currently represent the most comprehensive set of guidelines in systematic literature review research, ensuring the highest reproducibility and eligibility of the systematic review to ultimately provide novel insights. Considering that we are not able to make the tables according just one nutrient because of how the study was performed according to the eligibility criteria and inclusion and exclusion guidelines. It is a very good idea with some kind of signs (arrow up and down) as you suggest. We hope that the revised outcome columns in table 1-3 are easier to understand in the revised manuscript. Regarding your very valuable comment on clarifying which nutrients that is possible to conclude something from and which it cannot be done at present, we have included more on this in the beginning of the conclusion sections. We hope this will suffice. We have specified this point in line 320-323. "No prenatal, dietary factor has unequivocally been identified that modulate the frequency of T1D in the child, and no single, dietary factor can therefore, at the present, be recommended or discouraged from the maternal diet to minimize the disease development in a child with high genetic risk for T1D."

We have noticed that besides the written comments, you have stated that the presentation of the results and how the conclusions are supported by the results must be improved. We have therefore also added a section in the Conclusions section on the new insights or future directions to new research that we hope could improve this section. Our systematic review and its results show that data in the literature represents a heterogenous amount of data, meaning that new research deciphering the interplay between maternal dietary factors and the incidence of type 1 diabetes in the offspring needs to be performed. In addition, we have noticed that all reviewers find that some of the written material presented in the Discussion section might fit better in the Results section. This is, however, done according to the guidelines of PRISMA-P and PROSPERO. As we understand that you may also want us to provide some hypotheses on potential mechanisms by which maternal dietary factors relate to incidence of type 1 diabetes in the offspring, we have also included a section on this in the Discussion section. Thank you once again.

Reviewer 2 Report

The manuscript describes the results of a systematic review examining the association between maternal dietary factors during gestation and the risk of developing type 1 diabetes. The topic is interesting and the text is well written.

However, some points should be improved. First of all, the discussion section looks like «results» and could be moved there. Instead, please provide some hypotheses on potential mechanism of action of the foods shown to be associated with type 1 diabetes in the offspring.

Abstract

Please unfold IA when first used.

Methods

Please provide definitions of AI used. What were the criteria of IA?

The section «Data Collection Process» could be expanded.

Discussion

Please add some explanation for the role of maternal gluten consumption on the development of type 1 diabetes in the offspring.

 Line 191 The word «too» seems misleading in this context.

Lines 205-207 The sentence seems unfinished. Please check the meaning.

Table 1. The text in the column «Findings described relative to control group» is difficult to read and to distinguish between rows.

Author Response

Reviewer 2

The manuscript describes the results of a systematic review examining the association between maternal dietary factors during gestation and the risk of developing type 1 diabetes. The topic is interesting, and the text is well written. However, some points should be improved. First of all, the discussion section looks like «results» and could be moved there. Instead, please provide some hypotheses on potential mechanism of action of the foods shown to be associated with type 1 diabetes in the offspring. Abstract: Please unfold IA when first used. Methods: Please provide definitions of AI used. What were the criteria of IA? The section «Data Collection Process» could be expanded. Discussion: Please add some explanation for the role of maternal gluten consumption on the development of type 1 diabetes in the offspring. Line 191 The word «too» seems misleading in this context. Lines 205-207 The sentence seems unfinished. Please check the meaning. Table 1. The text in the column «Findings described relative to control group» is difficult to read and to distinguish between rows.

Response: Thank you very much for the comments. We have now inserted a paragraph in the text (line 302-308) providing some potential "mechanisms of action ". This is provided in line 307-313: “Potential mechanism of how diet during pregnancy could influence the development of IA and type 1 diabetes in the offspring could be the promotion of either an inflammatory or anti-inflammatory environment (references) affecting the child's developing immune system. Furthermore, maternal nutrition during pregnancy can potentially induce epigenetic changes in the child's DNA (references) or by the influence of the composition of the resident microbiota that is transferred to the child during a vaginal birth.” Thank you for suggesting more potential explanation of the role of maternal gluten consumption on the development of T1DM in the offspring. We have lines 249-250 in the revised manuscript that considers this and have added a more broad mode of action considering nutrients in general in line 317-323 in the revised manuscript. Thank you for pointing out the fact that we haven’t defined IA in the abstract. This is now done in the revised version of the manuscript. Parameters considered to assess islet autoimmunity IA, as you have noticed we missed in the first version of the manuscript, in the animal studies were: insulitis degree/score (infiltration of immune cells, mononuclear cell infiltration in islets). Where information of degree of apoptotic cells, inflammatory cell signaling, inflammatory marker RNA levels, and cytokine protein levels were included in the articles reviewed, this information was also provided in the tables. This information is now included in the manuscript in line 124 -128. Thank you for pointing out problems with the word “too” in the original draft line 191. This is now erased and changed to “also” at another place in the sentence (line 202 in the revised manuscript). Thank you for pointing out issues with line 205-207 in the original version of the manuscript. This is now changed in line 214-218 in the revised manuscript. Thank you for pointing out issues with table 1 and the specific column, we hope it is easier now in the revised version, where we have provided more space between rows.  

Reviewer 3 Report

Ingemann Johansen et al., performed a systematic review to summarize the current knowledge about dietary interventions in the origin or prevention of type 1 diabetes. The authors included articles carried out using animal models and clinical studies. After the exclusion criteria, 35 articles were included, mainly focused on the effects of vitamin D, gluten, fatty acid, and iron supplementation. The manuscript is well-written, with a comprehensive literature analysis, but the conclusions do not provide new insights or future directions to new research, which could improve this section.

There are some issues that the authors should address.

Define the range of dates that searching covers.

How did the authors define a relevant exposure, relevant study design, relevant outcome, and relevant study?

¿What parameters were considered to assess the islet autoimmunity in the animal studies?

Table 2 should include the origin of the populations of each study.

A section about the other dietary factors should be added to the discussion.

Minor comments,

In abstract, define “IA”

Author Response

Reviewer 3

Johansen et al., performed a systematic review to summarize the current knowledge about dietary interventions in the origin or prevention of type 1 diabetes. The authors included articles carried out using animal models and clinical studies. After the exclusion criteria, 35 articles were included, mainly focused on the effects of vitamin D, gluten, fatty acid, and iron supplementation. The manuscript is well-written, with a comprehensive literature analysis, but the conclusions do not provide new insights or future directions to new research, which could improve this section.There are some issues that the authors should address. Define the range of dates that searching covers. How did the authors define a relevant exposure, relevant study design, relevant outcome, and relevant study? ¿What parameters were considered to assess the islet autoimmunity in the animal studies? Table 2 should include the origin of the populations of each study. A section about the other dietary factors should be added to the discussion. Minor comments, In abstract, define “IA”.

Response: Thank you for the suggestions. We have now inserted a paragraph (line 335-338) regarding the lack of conclusions providing new insights or future directions. "Furthermore, there is a lack of concluding studies and review/ meta-analysis investigating the effect of non-medical interventions (for example, the role of psychical activity, diet, probiotics, sugar-containing beverages, psychosocial support) that could be used in IA positive children to prevent progression to clinical type 1 diabetes". Regarding definition of the range of dates that our systematic literature covers, this information is provided in line 75-76. We have hopefully made it more clear now by adding the extra information. The relevant exposure, relevant study design, relevant outcome and relevant studies are described according to PRISMA-P and PROSPERO. This information is included in the manuscript in line 90-103 (PICO elements), line 112-116 (study design), line 117-126 (eligibility criteria). Parameters considered to assess islet autoimmunity in the animal studies were: insulitis degree/score (infiltration of immune cells, mononuclear cell infiltration in islets). Where information of degree of apoptotic cells, inflammatory cell signaling, inflammatory marker RNA levels, and cytokine protein levels were included in the articles reviewed, this information was also provided in the tables. This information is now included in the manuscript in line 124 -128 . Thank you for this great suggestion. Regarding the suggestion about adding origin of populations of each study in table 2, this information is included in the “Subject type” column under “Sample”. Hope this information will suffice. We agree that it indeed is interesting with sections about the other dietary factors, however, we hope you will agree that given the very low amount of evidence investigating these nutrients and also considering the fact that much of the literature on these other nutrients comes from food frequency questionaries, it is more appropriate to conclude on this together as we have done in line 327-330. And thank you for pointing out the fact that we haven’t defined IA in the abstract. This is now done in the revised version of the manuscript.
